# Age-Related Conservation in Plant–Soil Feedback Accompanied by Ectomycorrhizal Domination in Temperate Forests in Northeast China

**DOI:** 10.3390/jof10050310

**Published:** 2024-04-24

**Authors:** Zhen Bai, Ji Ye, Shu-Fang Liu, Hai-Hong Sun, Zuo-Qiang Yuan, Zi-Kun Mao, Shuai Fang, Shao-Fen Long, Xu-Gao Wang

**Affiliations:** 1CAS Key Laboratory of Forest Ecology and Management, Institute of Applied Ecology, Chinese Academy of Sciences, Shenyang 110016, China; maozikun15@126.com (Z.-K.M.); fangs5@126.com (S.F.); long18310858051@163.com (S.-F.L.); wangxg@iae.ac.cn (X.-G.W.); 2College of Rural Revitalization, Weifang University, Weifang 261061, China; 20220058@wfu.edu.cn; 3Liaoning Provincial Institute of Poplar, Yingkou 115000, China; 15141849888@163.com; 4School of Ecology and Environment, Northwestern Polytechnical University, Xi’an 710072, China; zqyuan@nwpu.edu.cn; 5University of Chinese Academy of Sciences, Beijing 100049, China

**Keywords:** ectomycorrhizal (EcM) fungi, forest aging, soil nutrient dynamics, plant traits, temperate forest ecology, mycorrhizal exploration type

## Abstract

This study investigates the effects of forest aging on ectomycorrhizal (EcM) fungal community and foraging behavior and their interactions with plant–soil attributes. We explored EcM fungal communities and hyphal exploration types via rDNA sequencing and investigated their associations with plant–soil traits by comparing younger (~120 years) and older (~250 years) temperate forest stands in Northeast China. The results revealed increases in the EcM fungal richness and abundance with forest aging, paralleled by plant–soil feedback shifting from explorative to conservative nutrient use strategies. In the younger stands, *Tomentella* species were prevalent and showed positive correlations with nutrient availability in both the soil and leaves, alongside rapid increases in woody productivity. However, the older stands were marked by the dominance of the genera *Inocybe*, *Hymenogaster*, and *Otidea* which were significantly and positively correlated with soil nutrient contents and plant structural attributes such as the community-weighted mean height and standing biomass. Notably, the ratios of longer-to-shorter distance EcM fungal exploration types tended to decrease along with forest aging. Our findings underscore the integral role of EcM fungi in the aging processes of temperate forests, highlighting the EcM symbiont-mediated mechanisms adapting to nutrient scarcity and promoting sustainability in plant–soil consortia.

## 1. Introduction

Concerns have arisen about nutrient preservation during forest succession, as stand age is considered a critical source of variability in net ecosystem productivity and nutrient stocks [1,2]. The early-successional stage is predominated by pioneer tree species with acquisitive life-history traits, such as light-demanding nature and high photosynthetic capacity, usually characterized by large leaf area and high leaf nutrient contents [2,3]. The nutrient-rich plant materials can substantially contribute to soil nutrient stocks within the uppermost horizon of the forest ecosystem. Through rapid litter decomposition, these tree species always contribute to the sequestration of nutrients via aggregation and chemical bonding within the soil matrix. In contrast, shade-tolerant or climax species prevalent at the late-successional stage adopt conservative resource-use strategies and may experience decreases in net photosynthetic rates, primary production, and belowground carbon (C) and nutrient allocation [1,2,4,5,6,7]. Alongside forest aging, litter quality is also reduced, as observed through increased lignin/nitrogen (N) and C/N ratios, which further impede the decomposition rates and organic matter allocation to soil [7,8,9]. The aging of primary boreal conifer forests, some of which extending up to 560 years, is marked by the declined N availability in the soil [10]. N limitation promotes belowground C allocation to microbial communities, particularly ectomycorrhizal (EcM) fungi, facilitating N release from recalcitrant organic matter and aiding in its translocation via microbial networks [10,11]. The domination of microbial communities that favor organic matter utilization in older forests can lead to a reduction in microbial diversity due to nutrient depletion over the course of forest succession. Thus, forest aging is believed to result in the composition and functional convergence within soil microbial communities [12,13]. Nonetheless, the mechanisms underlying plant–soil–fungal dynamics and related nutrient use strategies remain largely ambiguous as forests age.

EcM fungi are closely related to vegetation community assembly and soil nutrient cycling dynamics during forest aging [6,14,15,16]. For instance, EcM fungi, in exchange for photosynthetic C compounds from hosts, enable better nutrient acquisition by increasing the accessibility of soil substances to tree roots, excreting many hydrolytic and oxidative enzymes and exhibiting high hydrolytic ability in scavenging primarily N and phosphorus (P)-containing organic compounds [17,18,19]. Well-established EcM fungal symbioses can exclude free-living fungal saprotrophs and subsequently monopolize organic matter decomposition and nutrient transfer to the belowground soil matrix [20,21,22,23,24,25]. Thus, a high abundance of ectomycorrhizae with emanating extraradical mycelia parallels slow-growing and nutrient-conservative plant traits (e.g., low foliar N content) and energy limitation in conservative niches [16,17,18,26]. EcM fungi employ different exploration strategies by extending various distances of extramatrical mycelia to soil [27,28]. These EcM exploration types include the contact exploration type with a smooth mantle and a few emanating hyphae (e.g., *Russula* and *Tomentella*); short-distance exploration type consisting of a voluminous envelope of emanating hyphae but without rhizomorphs (e.g., *Genea* and *Inocybe*); medium-distance exploration type forming rhizomorphs, which is further divided into the subtypes of medium-distance fringe (e.g., *Piloderma*, *Cortinarius*, and *Amphinema*) and medium-distance smooth (e.g., *Amanita*, *Lactarius*, and *Tricholoma*); and long-distance exploration type with smooth ectomycorrhizae with few but highly differentiated rhizomorphs (e.g., *Suillus* and *Rhizopogon*) [27]. It is believed that shorter-distance exploration types (e.g., contact, short-distance) may cost less in hyphal transition by host plants than their longer counterparts, and such distinctions may change the preference for substrate quality, including shorter exploration types for labile nutrient pools, and longer ones for complex and recalcitrant organic sources [29,30]. Recent studies have provided insights into the dynamics of EcM fungi in relation to forest succession. These studies have documented not only changes in the ratios of fungal exploration types but also the emergence of specific indicator species that are the characteristics of forests of different ages [25]. For instance, early successional-stage forests are highly abundant with cord-forming EcM basidiomycetes (*Suillus* and *Piloderma* species) [31]. However, old forest stands are more prevalent with shorter-distance exploration types and present a shift from the family Suillaceae to the family Russulaceae [32]. Despite differences in exploration distances or indicator taxa, similar distributing patterns of EcM fungal communities may be observed across forest chronosequences [33,34]. As such, understanding how EcM fungal communities and their functions change during forest aging is essential for comprehending forest succession and ensuring the long-term stability and productivity of forest ecosystems.

Northern temperate forests are highly dependent upon EcM symbioses because of low N availability and cold climates [35,36,37]. The broadleaf and Korean pine mixed forest on Changbai Mountain is one of the most important temperate forests and has a high aboveground C sequestration of more than 55 Mg C ha^−1^ [38,39]. The canopy dominants in these forest stands have experienced continuous regeneration for over two centuries, with research indicating that the C-carrying capacity of natural forests is typically reached at the 220-year mark [40,41]. To date, knowledge is limited about the age-related dynamics of the EcM fungal community diversity and composition driving above- and belowground nutrient use strategies, especially exploitative vs. conservative traits, across a chronosequence and among multiple tree species in the current forest ecosystems. Our study investigated soil properties, soil EcM fungi, vegetation composition and traits across younger (114–122 years) and older (247–251 years) stands of broad-pine mixed forests (140 subplots in total). We aimed to elucidate the intricate relationships among plant composition and traits, nutrient cycling dynamics, and EcM fungal communities during temperate forest aging. We hypothesized that the aging of temperate forests might lead to several changes: (1) an adaptation of plant community composition and functional traits toward less exploitative and more nutrient-conservative strategies, coupled with a prevalence of EcM symbionts; (2) a shift in EcM fungal community composition, potentially affecting the balance of exploration types covering various-distances; and (3) the emergence of indicator taxa of EcM fungi specific to different stages of forest succession. Our findings showed that temperate forest aging resulted in conservative traits along the plant–soil continuum, a high prevalence of EcM-associated trees and fungal communities, and the distinct dynamics of EcM fungal exploration types and indicator species.

## 2. Materials and Methods

### 2.1. Site Description

We conducted this research in mixed forest stands of broadleaf and Korean pine species, primarily composed of *Pinus koraiensis*, within six locations (124.91–127.98° E, 41.33–44.06° N) across the Jilin and Liaoning provinces in Northeastern China. These stands were categorized into the following two age groups: younger (114–122 years) and older (247–251 years). The tree age was estimated from the diameter at the breast height of 20 sample trees per forest stand. The climate of the study area has an annual average temperature ranging from 1.32 to 4.09 °C and precipitation between 666 and 951 mm. The soil type is identified as dark brown forest soil, according to the Food and Agriculture Organization of the United Nations classification [42]. Since 1998, these forests have been protected from human interference, allowing for natural succession from early to mature climax communities. Permanent vegetation plots measuring 0.6–1 hectare were established from July 2012 to July 2013, with further subdivision into nonoverlapping 20 m × 20 m subplots for a detailed study of plant and soil traits, ultimately comprising 140 subplots across all sites (15–20 replicates per site) [42].

### 2.2. Biotic Predictors of Vegetation Composition and Diversity

Between 2017 and 2018, we inventoried the plant life in each subplot, recording over 23,000 tree individuals from 81 species, 46 genera, and 26 families (diameters at breast height (DBH) > 1 cm) [42]. The stand basal area (Ba) was computed by summing the basal areas of all individual trees in each subplot [43]. The aboveground biomass (Biomass) of all the individual plants was calculated using allometric regression equations based on the diameter at breast height [44,45]. Tree species diversity was quantified through Shannon–Wiener species diversity (H) and tree species richness (S) in each subplot [45,46]. We identified the EcM-associated tree species according to the previous studies [47,48]. The proportions of EcM- and arbuscular mycorrhizal (AM)-associated trees were calculated based on their proportional basal area. We determined the functional traits of the dominant species, such as leaf area (LA), leaf P content (LPC), leaf N content (LNC) and maximum tree height (H) [46,49,50,51]. Furthermore, the community-weighted mean (CWM) of each functional trait, i.e., CWM.LA, CWM.LPC, CWM.LNC and CWM.H, was derived by weighting the relative Ba of the species in each subplot using the FD package in R [3,45,52]. The coarse woody productivity (CWP, Mg ha^−1^ year^−1^), which indicates the annual aboveground biomass increment from 2012 to 2018, was derived from the mean yearly aboveground growth of survivors and yearly recruitment of individuals (DBH > 1 cm) [45].

### 2.3. Soil Physicochemical and Bioinformatic Analyses

In August 2018, we collected composite soil samples by thoroughly mixing five randomly selected soil cores (0–10 cm) from each 20 m × 20 m subplot. Subsequently, the samples were cooled to 4 °C for transport, sieved (<2 mm), stored at −80 °C for DNA sequencing or air-dried for chemical assessment. As previously reported [46,48], we evaluated the soil chemical properties, including soil organic C (SOC), total N (TN), total P (TP), total potassium (TK), readily available N (AN) and P (AP), and pH.

Genomic DNA was extracted from 0.25 g of soil via the MoBio PowerSoil^®^ DNA Isolation Extraction Kit (MoBio, Carlsbad, CA, USA). The DNA quality was evaluated through absorbance ratios at 260/280 nm and 260/230 nm, using a NanoDrop Life Spectrophotometer (NanoDrop, Wilmington, DE, USA). The high-quality DNA was then preserved at −40 °C for subsequent analyses. Amplification of the internal transcribed spacer (ITS) region of the 18S rRNA gene was performed using the primers ITS1 (5′-CTTGGTCATTTAGAGGAAGTAA-3′) and ITS2 (5′-GCTGCGTTCTTCATCGATGC-3′) [12]. We utilized the GeneAmp PCR System 9700 (Applied Biosystems, Foster City, CA, USA) for the polymerase chain reactions (PCRs). Sequencing was then conducted on a 300 PE MiSeq platform (Illumina, San Diego, CA, USA) by Shanghai Majorbio Biopharm Biotechnology Co., Ltd. (Shanghai, China).

Subsequent data processing was performed with QIIME (version 1.7). Sequence reads were quality-filtered, trimmed, and assigned to their respective samples according to their barcodes. Only high-quality sequences characterized by a length greater than 200 base pairs, no ambiguous bases, and an average base quality score greater than 25 were considered for further analyses. The sequences were grouped into operational taxonomic units (OTUs) at a 97% identity threshold with UPARSE (version 7.1), while singletons and doubletons were excluded. The 18S rRNA sequences underwent a chimera check utilizing the UCHIME algorithm. Normalization of the sequencing depth was achieved by subsampling each sample to 23,902 reads for 18S rRNA. We assigned taxonomies with representative sequences from the UNITE7.0 database. For the assessment of fungal community diversity within the samples, alpha diversity metrics, including the Shannon–Wiener diversity and Chao1 indices, were calculated using QIIME’s “core_diversity_analyses.py” script. The Chao1 and Shannon indices measure the richness and uniformity of the species present, respectively [53]. FungalTraits was used to assign the EcM fungal traits [54,55].

### 2.4. Statistical Analyses

The influences of forest aging on EcM fungal alpha-diversity, exploration type, and indicator species were evaluated as follows. The normality and variance homogeneity of the data were assessed using Shapiro–Wilk and Levene’s tests, respectively. For normally distributed data, the differences were analyzed using one-way analysis of variance (ANOVA) with post hoc comparisons via Tukey’s honestly significant difference (HSD) test. Data not meeting the normality assumption were analyzed using the Kruskal–Wallis test, a nonparametric alternative. The raw data were standardized before further analyses. The analyses were executed using the statistical functions available in R software (R Development Core Team, 2022, version 4.1.3). Statistical significance was defined as *p* < 0.05. We performed Spearman’s correlation analysis on variables to assess correlations via the ‘corrr’ package in R. We carried out nonmetric multidimensional scaling analysis (NMDS) to explore the differences in the EcM fungal community in relation to forest stand age. Analysis of similarity (ANOSIM) and permutational multivariate analysis of variance (PERMANOVA) revealed the influence of the aging effect with Bray–Curtis similarities and 999 permutations. PERMANOVA relies on permutation tests and can be used for data not fitting the normality assumption [56]. In addition, the random forest classification approach, a nonlinear and multivariate method known for its robustness in handling cluster-based data, was selected to discern specific tree and EcM fungal species associated with either younger or older forest stands. This analysis was performed using the ‘randomForest’ package (version 4.7-1.1) in R. To further validate the importance of predictor variables in the model, we employed VSURF within the same ‘randomForest’ package.

## 3. Results

### 3.1. Plant–Soil Attributes

Significant changes were observed with the stand basal areas (Ba) of various tree species as the forests progressed through different stages (Figure 1, *p* < 0.05). The earlier successional stage predominantly featured tree species such as *Ulmus laciniata*, *Acer mandshuricum*, *Euonymus phellomanus*, *Syringa reticulate*, *Phellodendron amurense*, *Abies holophylla*, *Acer mono*, and *Actinidia kolomikta*. Conversely, the older stands exhibited greater Ba for the tree species *Acer pseudosieboldianum*, *Acer rubripes*, *Abies nephrolepis*, *Picea jezoensis*, *Acer barbinerve*, *Pinus koraiensis*, *Tilia amurensis*, and *Quercus mongolica*. A notable contrast was seen between *Acer pseudosieboldianum* and *Ulmus laciniata*, where the Ba of the former markedly increased by more than seven times with aging, while the latter declined its Ba by approximately 87% in the older stands relative to the younger stands.

There was a noticeable shift in the community-weighted mean (CWM) traits associated with tree growth and ecological function. Specifically, the CWM of maximum tree height (CWM.H), Ba, Biomass, and the proportion of EcM-associated trees significantly increased by 16–55% with increasing forest age (*p* < 0.05). These increases reflect the trend toward more conservative growth strategies with forest aging. In stark contrast, the CWM values of the leaf area (CWM.LA), leaf N content (CWM.LNC), and leaf P content (CWM.LPC), as well as plant species richness (S) and diversity (H), showed substantial increases of 11–200% in the younger stands compared with the older stands (*p* < 0.05). These patterns align with exploitative strategies, characterized by rapid growth and a high demand for nutrients. Together, our findings reflect the trend from fast-growing and exploitative to conservative tree species with the aging of temperate forests.

Soil analyses revealed that nutrient availability markedly declined as the forests aged. The contents of AP, TC, and TN displayed the most significant differences, with reductions of 147–200%, when comparing younger to older stands (*p* < 0.05). Our results elucidate a successional trajectory in current temperate forests from younger, nutrient-rich, and species-diverse stands with traits favoring rapid growth and resource use (exploitative strategy) to older, nutrient-conserving forests characterized by greater biomass and tree height (conservative strategy). These findings highlight the clear transition from exploitative to conservative ecological strategies as forests mature.

### 3.2. EcM Fungal Diversity

Alongside the shifts in plant–soil dynamics, significant increases in EcM fungal diversity and abundance were observed as the forests aged (Table 1). EcM fungal richness varied from 53.82 to 63.20 in the younger stands to 67.40 to 100.80 in the older stands, exhibiting a marked upward trend (35% increase) during forest succession (*p* < 0.001). Diversity estimators, including the Chao1 estimator (*p* < 0.001) and abundance-based coverage estimator (Ace) (*p* < 0.05), exhibited average increases of 13–20% as the forests aged. These estimators pointed to a broader range of EcM fungal diversity in the older stands, with values increasing from 85.19 to 111.69 in the younger stands to 88.18 to 133.27 in the older stands. The Good’s coverage index approached unity in the older stands and its values were approximately 4% greater than those in the younger stands (0.95–0.97) (Table 1, *p* < 0.001). This robustness of the data implies that the estimates of EcM fungal richness were reliable and encompassed most of the actual diversity presented.

Further analysis of the EcM fungal community structure revealed a significant differentiation associated with forest aging. ANOSIM showed a global R of 0.542 (*p* < 0.001) and PERMANOVA had an R^2^ of 0.082 (*p* < 0.01) (Figure 2). Both sets of statistics confirmed significant divergence in community composition, indicating that the structure of the EcM fungal communities was indeed influenced by the aging of the temperate forests. In summary, our findings revealed a clear shift in EcM fungal diversity with forest aging. This trend was consistent across various diversity measures and was further corroborated by near-complete sampling in current forest stands. The significant divergence in the EcM fungal community composition between the younger and older forests underscores the impact of forest succession on the EcM fungal biodiversity and community dynamics.

### 3.3. EcM Fungal Exploration Types

The EcM fungal abundances associated with the different exploration types displayed varying responses to forest aging, with a minimal increase in the long-distance type (a 0.17-fold rise) and a maximal increase in the short-distance coarse type (a 5.92-fold escalation) (Table 2, *p* < 0.001). Of all the foraging patterns, only the contact type presented a consistent and uniform aging effect within both the younger and older age groups. In the younger stands, the contact type made up 3.36–5.22% of the relative abundance, which surged to 9.62–19.25% in the older stands, averaging a 2.38-fold increase (*p* < 0.001).

The ratios of longer-to-shorter distance exploration types either decreased significantly or remained steady with forest aging. These trends revealed that the abundance of EcM fungi associated with shorter-distance exploration types might outweigh those associated with longer-distance types as the forests aged. For instance, the ratio of long-to-contact on average declined by 0.96 times from the younger to older stands (*p* < 0.001). The most notable decline was a 0.99-fold reduction, observed by pairwise site comparison between ltd (5.70) and xp (0.03). No aging effect was observed on the ratios of middle-to-short distance exploration types (*p* > 0.05).

Moreover, the associations of the EcM fungal exploration types with plant–soil traits varied with forest age (Figure 3). In the younger stands, the EcM fungi of the contact exploration type presented no correlation with any soil variable (*p* > 0.05); however, in the older stands, they were significantly positively related to the soil C/P ratio (*p* < 0.05). Similarly, plant traits such as richness (S) had no correlation with any of the exploration types of EcM fungi in the younger stands (*p* > 0.05) but developed strong positive correlations with contact, short-, and medium-distance exploration types in the older stands (*p* < 0.05). Moreover, the younger and older stands displayed opposing trends regarding soil C/P ratio and N/P ratio when correlated with the EcM fungal exploration type—negative in the younger stands versus positive in the older stands (*p* < 0.05). Further, strong relationships (*p* < 0.05) unique either to the younger or older stands were observed between EcM fungi with distinct EcM exploration types and tree species, such as *Pinus koraiensis*, *Tilla amurensis*, *Acer rubripes*, *Phellodendron amurense*, *Euonymus phellomanus*, and *Actinidia kolomikta*.

Despite the abovementioned differences in aging, certain relationships between plant traits, soil attributes, and EcM fungal exploration types were consistent across forests of different maturities. For instance, CWM.H and EcM tree proportions were positively related to the long-distance, medium-distance fringe (mdf), and/or short-distance coarse (sdc) types in both the younger and older age groups (Figure 3, *p* < 0.05). Additionally, the nutrient contents in leaf and soil generally presented negative correlations with EcM fungi of different exploration types irrespective of forest age (*p* < 0.05), e.g., inverse relationships between soil nutrients and various EcM types.

### 3.4. Age-Differentiated EcM Fungal Species

We identified 65 EcM fungal species that were instrumental in differentiating between the early and late forest stands using the ‘randomForest’ package in R (Figure 4, *p* < 0.05). Among them, 23 exhibited weak associations with plant–soil traits and were thus removed from subsequent analyses. The refined list comprised 42 species (Figure 5, Table 3).

In the younger stands, specific *Tomentella* species (namely, *Tomentella*5, *Tomentella*51, and *Tomentella*63) demonstrated consistent rises in the relative abundance (Table 3, *p* < 0.001). These species showed positive correlations with nutrient contents (e.g., TC, TN, AN, or AP) and plant explorative traits, especially CWM.LPC (Figure 5, *p* < 0.05). *Sebacina* species (e.g., *Sebacina*12), another taxon that predominated in the younger stands (Table 3, *p* < 0.001), was positively related to Biomass and Ba (Figure 5, *p* < 0.05).

However, the older forest stands were more abundant with the fungal genera *Gene* (*Genea*7), *Rhizopogon* (*Rhizopogon*1), *Inocybe* (*Inocybe*2, *Inocybe*29), *Russula* (*Russula*3, *Russula*23), *Suillus* (*Suillus*13), *Piloderma* (*Piloderma*41), and *Amanita* (*Amanita*52) (Table 3, *p* < 0.001). In addition, the species *Tylospora*38 and *Lactifluus*57 were positively and significantly correlated with CWM.H and CWP (Figure 5, *p* < 0.05). Furthermore, the abundances of the species *Inocybe*56, *Hymenogaster*22, *Pseudotomentella*54, *Otidea*25, and *Tomentella*39 had notable positive correlations with the contents of soil TN, TP, and AN, and with the values of CWM.LPC and CWM.LNC (*p* < 0.05). Moreover, the *Genea* (*Genea*31) and *Clavulina* (*Clavulina*6) species displayed significantly positive relationships with plant diversity indices, such as the Shannon diversity index (H) and species richness (S) (*p* < 0.05).

## 4. Discussion

This study investigated the dynamics of edaphic properties, plant traits, and EcM fungal dimensions prompted by an aging process spanning approximately 130 years in broad-leaved and Korean pine mixed forests in Northeast China. Our aim was to uncover the important roles of EcM fungi, especially the exploration types and key indicator taxa, in plant–soil interactions during forest aging. We demonstrated that age-associated conservative traits and irreversible reductions in nutrient cycling dynamics along the plant–soil continuum were closely related to the predominance of EcM fungi.

### 4.1. Conservative Plant–Soil Traits Coupled to EcM Fungal Dominance

Forest aging caused a marked transition in tree community dynamics, especially between *Ulmus laciniata* and *Acer pseudosieboldianum*. These species are frequently encountered in natural broad-leaved and Korean pine (*P. koraiensis*) mixed forests [54,55]. The broadleaf tree *Ulmus laciniata* hosts EcM and AM fungi [47] and is an intermediate shade-tolerant species [56]. The increasing dominance of this species may be more efficient in mobilizing organic and recalcitrant nutrients, which in turn enhance the evenness and richness of tree species. *Acer pseudosieboldianum*, on the other hand, can be predominant in the later stages of mixed coniferous and deciduous broad-leaved forests because it is more shade tolerant and prevails in lower tree strata (e.g., understory and shrub layers) [57,58,59]. This shift in species dominance during forest succession strongly suggests a developmental trend from early-successional, fast-growing, exploitative species to late-successional, conservative species with forest aging. Corresponding to such tree species succession with aging, we observed notable shifts from exploitative dynamics, characterized by high values of CWM.LNC, CWM.LPC, and CWP, to conservative traits of high biomass accumulation and CWM.H. This exploitative-to-conservative transition was accompanied by a declining trend in soil nutrient availability (i.e., TC, TN, AN, and AP). Previous studies have shown that forest aging may have a more efficient internal circulation of N, reducing the photosynthetic function and net growth and belowground nutrient exhaustion; in turn, forest aging is characterized by increasingly recalcitrant polyphenolic compounds, decreased C use efficiency, less dependence upon nutrient availability, and a preference for fungal symbionts that require low nutrient investment [6,7,10,16,31].

In late-successional forest stands, where soil nutrients are less available, EcM fungi may increase their diversity and abundance to exploit the challenging conditions needed to break down more complex nutrient compounds [31,60]. Such a high predominance of EcM symbionts favors nutrient use-conservative strategies that downregulate leaf nutrient contents (e.g., LNC, LPC) and cause low decomposition rates and less belowground nutrient translocation [22,26,61,62]. Low vegetation diversity occurred in the EcM-dominated older forest stands (Figure 1), as thriving EcM symbionts often squeeze habitats for other biological processes and restrict the variety of woody vegetation [22,63]. Consequently, EcM dominance may sustain fewer simultaneous multifunctions and lower the richness of the soil microbiome. This low diversity under EcM dominance is accompanied by conservative traits and relatively low N and P use economies (e.g., low nutrient status in leaves and soil). Such a transition can negatively influence the consumption of available nutrients (i.e., AN and AP) and preserve valuable nutrients in the soil, leading to a high reliance of host plants on EcM symbioses at the late-succession stage of temperate forests. In summary, our findings corroborate our first hypothesis, demonstrating a pronounced shift from exploitative to conservative ecological strategies in both plant and fungal communities with forest aging. This pivot is intricately linked to the growing dominance of EcM fungi, which are well suited to the nutrient-conservative conditions of aged forests.

### 4.2. The Impacts of Aging on EcM Fungal Exploration Strategies

In the older stands, the prevalence of contact, short- and medium-distance exploration types was positively associated with the soil C/P ratio and tree richness but had a strong negative association with the CWM.LPC (Figure 3). It is believed that the EcM short-distance exploration type invests little in transporting mycelia and thus reduces photosynthesis investment and is more adaptable to conservative lifestyles compared to its longer-distance exploration counterparts [16,29,31,64]. The late-stage preference for short distances suggests that low-C symbiotic partnerships tend toward C retention and conservative nutrient (e.g., P) use strategies. This is reasonable because conservative nutrient economics are always accompanied by declines in photosynthesis and belowground C investment for nutrient exchange [5,6,65]. In addition, the canopy height (CWM.H) of the older stands seemed to be strongly related to the growth of the medium-distance fringe type, probably because these genera, such as *Tricholoma*, *Cortinarius*, and *Piloderma*, possess a greater capacity for exploring nutrients in organic-poor soil than in organic-rich soil due to their dense rhizomorphs [29,66,67]. This was strongly evidenced by the highly negative correlations between medium-distance fringe type and soil pH, and between medium-distance fringe type and soil TP, in the older forest stands, as P mobilization and acquisition are always driven by the production of organic acids [68,69]. Together, these findings explain why the medium-distance and short-distance types had similar responses to the aging process. Consequently, a decreasing trend in the ratio of longer-to-shorter distance exploration types occurred as the current forests aged. Despite inconsistent mycelial foraging strategies for exploration types [67,70], the EcM exploration distance types, especially the medium and short types, can serve as indicators of the aging process in temperate forests.

Moreover, the CWM.H in the older forest stands was remarkably influenced by fungi from the long- and medium-distance exploration types. Fast growth rates of stems are sustained by highly available canopy photosynthates produced under optimal environments (warm, wet, high soil quality) associated with light competition [71]. Such optimal conditions also cause intense competition for soil nutrient exploration. In contrast to shorter exploration types, long-distance exploration strategists may have greater hydrophobic rhizomorphs and water-holding capacity [72], are more effective in utilizing recalcitrant nutrients (P compounds) in soil [73], and can consistently increase N uptake by host plants under conditions of heterogeneity and nutrient deficiency [29,74]. Most importantly, long-distance EcM fungi, such as the *Suillus* species, can rapidly produce diffusing mycelia to colonize the soil matrix and be opportunistic in nutrient-use strategies [33]. These long-distance traits may be ascribed to the vast and far-reaching mycelia that aid in nutrient access, absorption, and transfer to well-developed host roots in old-growth forests [67]. Our findings imply that the vegetation height growth at later successional stages should depend, at least to a greater extent, upon retaining the capacity for a consistent and steady nutrient supply via long-distance EcM fungi. Combining the conservative traits of the plant–soil continuum, our second hypothesis is supported by the fact that the EcM fungal foraging traits are strongly determined by soil nutrient status and host nutrient demands.

### 4.3. EcM Indicator Species across Forest Successional Stages

The indicator taxa for the younger stands included the EcM fungal genera *Tomentella* and *Sebacina* (Table 3). The *Tomentella* species are very common in early colonizers of roots and are favored at nutrient-rich sites [33,75,76]. In the present study, the *Tomentella* species demonstrated positive correlations with soil TC, TN, TP, AN, AP, as well as LPC (Figure 5). In addition, the *Sebacina* species are typically associated with short-distance exploration and are frequently observed and predominant in early-stage forest stands, probably because they build widespread belowground networks commonly shared by various tree species [77]. Their presence can facilitate C and nutrient transfer between different host plants. This probably explains why *Sebacina* (*Sebacina*12) was highly correlated with the richness, biomass, and Ba of plant communities.

The older forest stands were more abundant with the EcM fungal genera *Genea* and *Russula* (Table 3), which exhibited strong positive associations with plant richness/diversity (H/S), CWM.H, and/or soil properties such as the TC/TN and TC/TP ratios (Figure 5). Old-growth forests, with their substantial aboveground biomass and great-height trees, rely on a steady nutrient supply from mycorrhizal associations and are more dependent on C fixed through photosynthesis, which is then allocated to root-associated EcM fungi in exchange for nutrients from decomposed organic matter. This symbiotic relationship underscores the importance of EcM fungi in the transition from longer to shorter mycelial exploration types observed during forest aging. For instance, *Genea* comprises the hypogeous Ascomycota EcM species (humus saprobic) and exhibits a short-range exploration strategy with a voluminous envelope of emanating hyphae but without rhizomorphs [27,78,79]. The *Genea* species can be key nodes with the greatest connectivity and complex mycorrhizal interactions with a wide variety of putative EcM host plants such as *Abies*, *Larix*, *Pinus*, *Tsuga*, *Pseudotsuga*, *Betula*, *Fagus*, *Quercus*, *Carpinus*, *Cistus*, *Nothofagus*, *Lithocarpus*, and *Corylus* [80]. Furthermore, *Russula* is generally characterized by short range of mycelium with smooth mantle and few emanating hyphae, is highly positively related to β-1.4-N-acetyl-glucosaminidase (N mobilization) and/or acid phosphatase (P mobiliza-tion) utilizing complex nutrient pools, and thus competitively prevails in the intensified nutrient limitation niche in later successional forest stands [25,33,76,81,82,83]. *Russula paludosa*, in particular, is regarded as an indicator of cool sites (i.e., those where the mean annual temperature is below 8–9 °C) [84].

Moreover, the *Amanita* species form far-reaching mycelia (medium-distance smooth), are able to access and utilize soil organic N (e.g., protein N) [85], and preferentially occur in low-fertility forest stands [86]. Usually, the *Amanita* species cooccur with mature trees and are frequently highly abundant in late-successional forest stands [25,33,75]. Furthermore, the *Suillus* species are affiliated to the long-ranging exploration type, and secrete β-1.4-N-acetyl-glucosaminidases and release N from recalcitrant organic matter (chitin), allowing the adaptation to nutrient-poor conditions [21,29,87,88]. Moreover, the long-distance exploration genus *Rhizopogon* is adept at translocating water and nutrients via an extensive mycelial network; this genus can occasionally trigger host expansion with other tree species [62,89]. We currently observed a significant positive correlation between *Rhizopogon* and CWM.H (Figure 5). This combination of traits of *Rhizopogon* partially explains why the long-distance type as a whole is highly correlated with CWM.H in old-growth forest stands. Together, the older stands were highly abundant with Suilloid fungi (*Suillus* and *Rhizopogon* species), probably due to these EcM fungal species’ long-distance mycorrhizal exploration capability and their particular adaptability to nutrient-deficient soils [84,90].

In summary, our study identified distinct EcM fungal taxa that were associated with different forest ages. The *Tomentella* and *Sebacina* species were associated with nutrient-rich conditions and high productivity in the younger forest stands. As the forests aged, the *Genea*, *Amanita*, *Russula*, and *Suillus* species became more prevalent, fulfilling roles that bolstered the productivity and nutrient retention of the plant–soil continuum. These findings confirm our third hypothesis, which postulated that there would be a succession of specific EcM fungal taxa indicators throughout forest development as follows: EcM fungi regulate the nutrient dynamics, with younger forests displaying a positive association with fungi that enhances nutrient availability and older forests showing an intricate balance that favors nutrient retention.

## 5. Conclusions

Our findings reveal a clear age-related succession in the temperate forests of Northeast China, marked by a transition from aggressive, growth-oriented strategies toward more conservative, resource-efficient traits. This succession is accompanied by the exploitative-to-conservative shifts in tree species, decreases in soil nutrient availability, and a downregulated leaf nutrient content and productivity, compared to biomass accumulation and mean height, pointing to a maturation favoring nutrient retention and stability. This transformation aligns with the corresponding increases in the EcM fungal richness and abundance. The balance of EcM fungal exploration types evolves with forest age, with long-distance exploration types becoming less dominant relative to short- and medium-distance exploration types. This shift further reflects the complex interplay between nutrient cycling, soil nutrient availability, and plant–fungal interactions, which are crucial for maintaining highly efficient nutrient uptake and retention by host trees. The identification of EcM fungal taxa indicators associated with different stages of succession also highlights a shift in fungal strategies from those that favor rapid nutrient uptake to those that support sustained nutrient cycling and stability within the ecosystem. In the younger stands, the genera such as *Tomentella* and *Sebacina* are correlated with nutrient cycling and productivity, while in the older forests, the genera such as *Genea*, *Amanita*, *Ru*ss*ula*, and *Suillus* emerge as significant contributors to the maintenance of productivity and a conservative nutrient-use economy. The intricate relationships between plant composition and traits, EcM fungal diversity and functions, and nutrient cycling dynamics are fundamental to forest development and have significant implications for the forest management practices aimed at promoting biodiversity, resilience, and ecosystem services.

## Figures and Tables

**Figure 1 jof-10-00310-f001:**
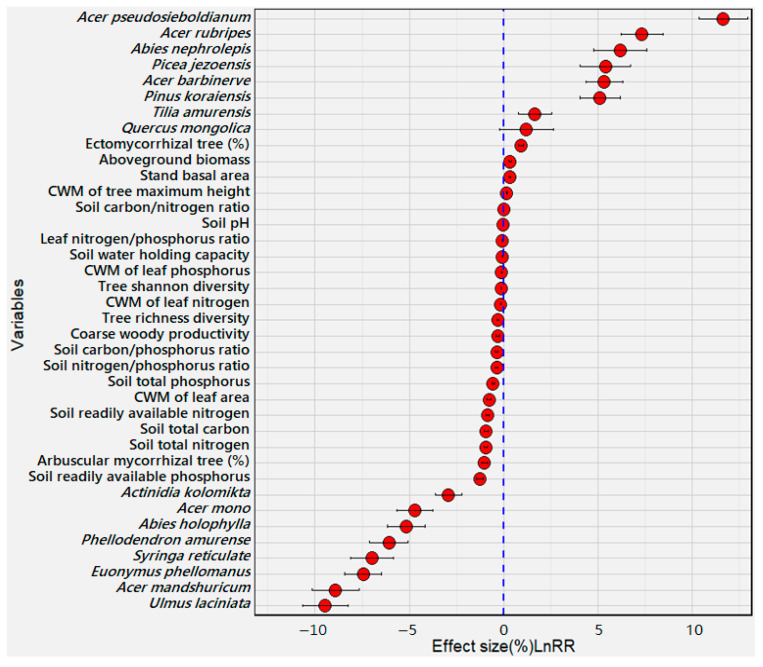
Effects of forest aging on the changes in tree species composition, plant traits, and soil properties. lnRR denotes the natural logarithm-transformed response ratio of each variable in the older forest stands against the younger forest stands. Filled circles with error bars represent the mean change in variables and their respective 95% CIs (confidence intervals). The dashed line indicates no aging effect. When the mean changes are greater than 0 (blue dashed line), the variables are more abundant in the older stands than in the younger stands; otherwise, the variables are more abundant in the younger stands compared with the older stands. When the 95% CIs do not overlap 0, the variables significantly differ between the younger and older stands. The community-weighted mean, CWM.

**Figure 2 jof-10-00310-f002:**
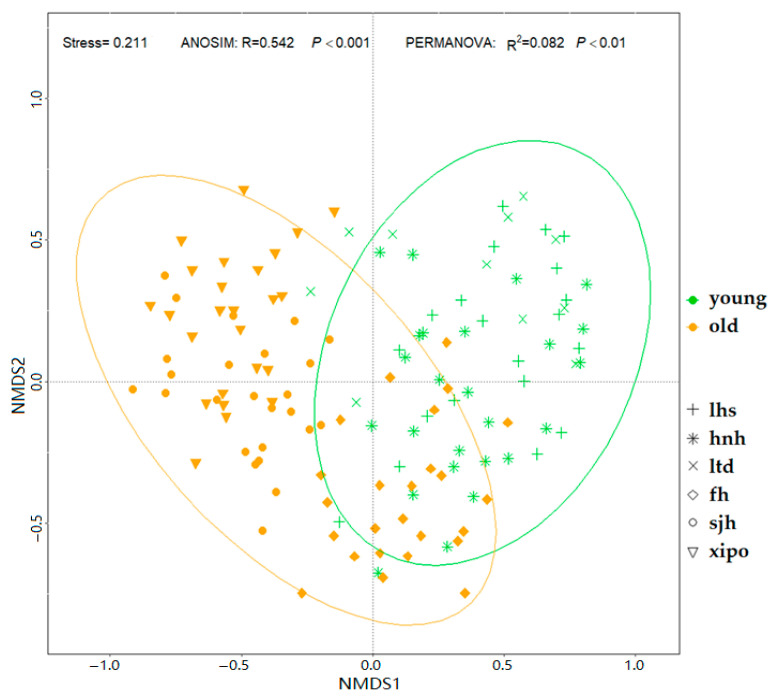
Nonmetric multidimensional scaling analyses of the EcM fungal community based on Bray–Curtis distances. Analysis of similarities (ANOSIM) and permutation multivariate analysis of variance (PERMANOVA, 999 permutations) reveals significant differences in the structure of the tree communities between younger and older forest stands. The younger and older stands are denoted by green and yellow, respectively. The symbols represent the different sampling plots (i.e., points that share a symbol are from the same site).

**Figure 3 jof-10-00310-f003:**
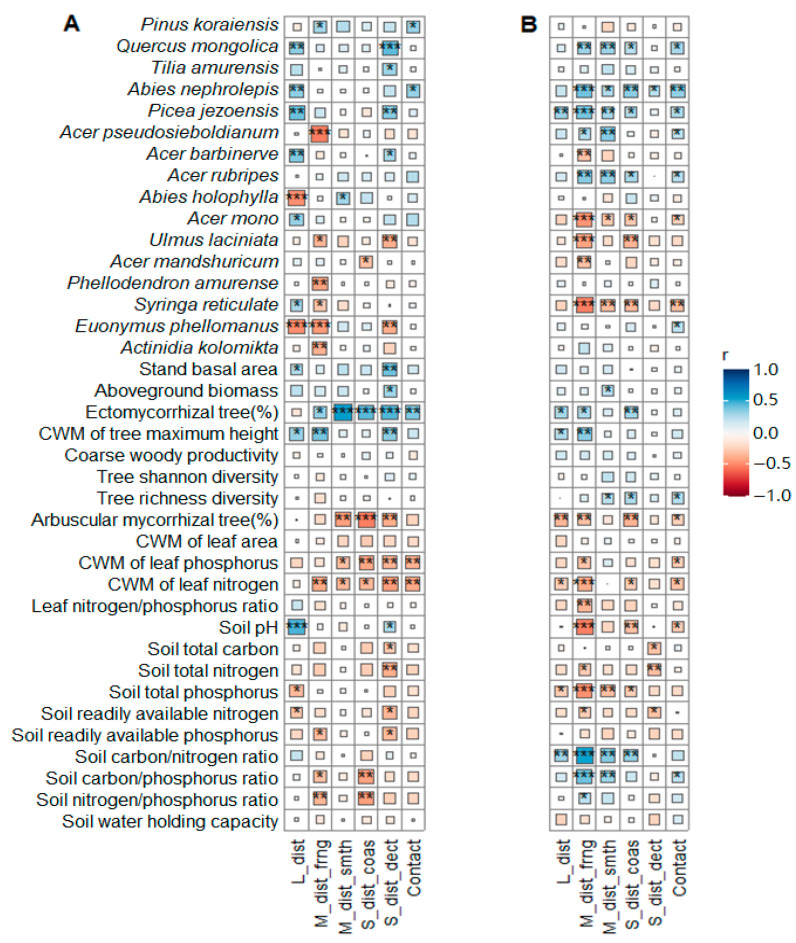
Correlations between EcM fungal exploration types and tree species, plant traits and soil properties in the younger (**A**) and older (**B**) stands. Spearman’s correlation analyses were performed on standardized variables to assess correlations via the ‘corrr’ package in R. Squares with blue and red colors indicate positive and negative relationships, respectively. Square size indicates the values of the correlation coefficients. Significant differences are denoted by asterisks: *, *p* < 0.05; **, *p* < 0.01; and ***, *p* < 0.001. The community-weighted mean, CWM.

**Figure 4 jof-10-00310-f004:**
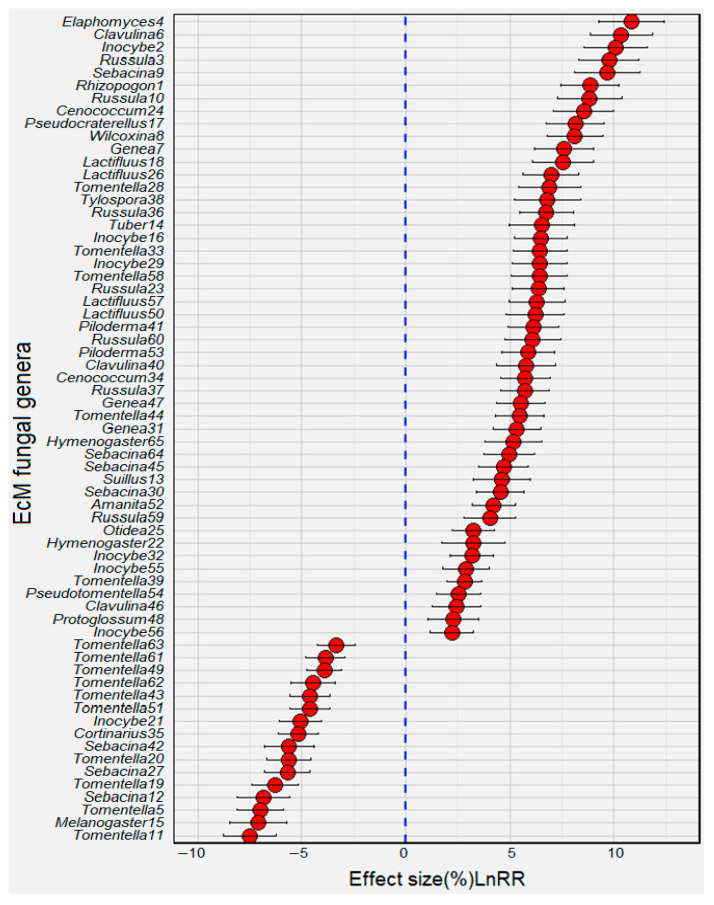
Effects of forest aging on the changes in the abundance of EcM fungal species. lnRR denotes the natural logarithm-transformed response ratio of each fungal species in the older forest stands against the younger forest stands. Filled circles with error bars represent the mean change in variables and their respective 95% CIs (confidence intervals). When the mean changes are lower than 0 (blue dashed line), fungal species are more abundant in the older stands than in the younger stands; otherwise, variables are more abundant in the younger stands than in the older stands. When the 95% CIs do not overlap 0, the variables significantly differ between the younger and older stands.

**Figure 5 jof-10-00310-f005:**
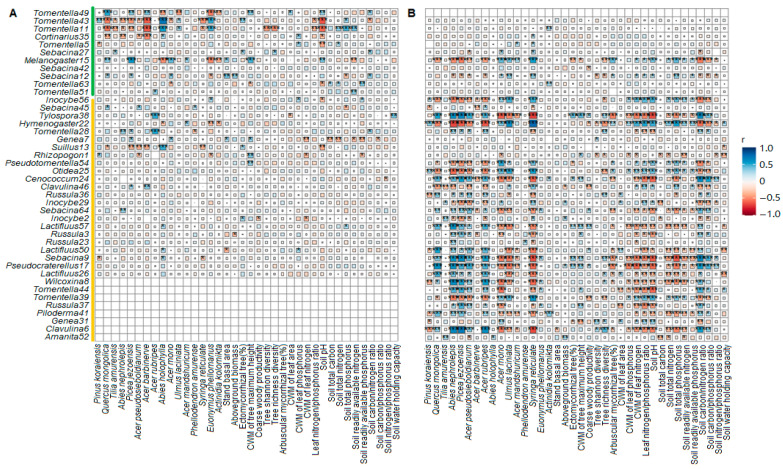
Correlations between EcM fungal species and tree species, plant traits and soil properties in the younger (**A**) and older (**B**) stands. Spearman’s correlation analyses were performed on variables to assess correlations via the ‘corrr’ package in R. Squares with blue and red colors indicate positive and negative relationships, respectively. Square size indicates the values of the correlation coefficients. Significant differences are denoted by asterisks as follows: *, *p* < 0.05; **, *p* < 0.01; and ***, *p* < 0.001. The community-weighted mean, CWM. Green and yellow bars refer to fungal species that are more abundant in the younger and older stands, respectively.

**Table 1 jof-10-00310-t001:** Alpha-diversity of soil EcM fungi in the younger and older forest stands. The values are presented as the means ± standard errors. The different characters in a single row indicate significant differences across the six sites at *p* < 0.05. In the column of (O-Y)/Y, significant differences between the younger and older stands are denoted by asterisks: *, *p* < 0.05; and ***, *p* < 0.001; values denote increases (folds) from the older stands (O) to the younger stands (Y); NS indicates no significant difference between the younger and older stands. The differences are analyzed using one-way ANOVA. When the data do not meet the normality assumptions, they are analyzed using nonparametric Kruskal–Wallis tests.

Alpha-Diversity	Younger Stands lhs (114 y, *n* = 25) hnh (117 y, *n* = 25) ltd (122 y, *n* = 15)			Older Stands fh (251 y, *n* = 24) sjh (247 y, *n* = 25) xp (250 y, *n* = 25)			(O-Y)/Y
Shannon	2.40 ± 0.17 a	2.22 ± 0.14 a	1.93 ± 0.20 a	2.20 ± 0.11 a	2.30 ± 0.11 a	2.41 ± 0.12 a	NS
Simpson	0.77 ± 0.04 a	0.75 ± 0.03 a	0.67 ± 0.05 a	0.78 ± 0.03 a	0.78 ± 0.03 a	0.78 ± 0.03 a	NS
Richness	63.20 ± 2.79 bc	71.72 ± 3.54 b	53.82 ± 5.98 c	67.40 ± 2.92 bc	95.32 ± 4.85 a	100.80 ± 3.29 a	0.35 ***
Chao1	92.24 ± 4.83 b	103.29 ± 5.05 b	85.19 ± 8.33 b	88.18 ± 4.08 b	129.21 ± 6.83 a	126.74 ± 4.41 a	0.20 ***
Ace	99.41 ± 5.06 cd	111.69 ± 5.46 bc	92.64 ± 7.59 cd	90.74 ± 4.15 d	133.27 ± 7.33 a	128.05 ± 4.33 ab	0.13 *
Pielou evenness	0.40 ± 0.03 a	0.36 ± 0.02 a	0.34 ± 0.03 a	0.36 ± 0.02 a	0.35 ± 0.02 a	0.36 ± 0.02 a	NS
Goods’ coverage	0.95 ± 0.01 c	0.97 ± 0.01 c	0.97 ± 0.01 c	0.99 ± 0.00 b	1.00 ± 0.00 ab	1.00 ± 0.00 a	0.04 ***

**Table 2 jof-10-00310-t002:** Soil EcM fungal types as affected by forest aging. Values are indicated as mean ± standard error. The different characters in a single row indicate significant differences across the six sites at *p* < 0.05. In the column of (O-Y)/Y, significant differences between the younger and older stands are denoted by asterisks as follows: *, *p* < 0.05; **, *p* < 0.01; and ***, *p* < 0.001; values denote increases (folds) from the older stands (O) to the younger stands (Y); NS indicates no significant difference between the younger and older stands. The differences are analyzed using one-way ANOVA. When the data do not meet normality assumptions, they are analyzed using nonparametric Kruskal–Wallis tests. The exploration types include the contact, short-distance coarse (S_dist_coas), short-distance delicate (S_dist_dect), medium-distance fringe (M_dist_frng), medium-distance smooth (M_dist_smth), and long-distance (L_dist) types.

EcM Types (%)	Younger Stands lhs (114 y, *n* = 25) hnh (117 y, *n* = 25) ltd (122 y, *n* = 15)			Older Stands fh (251 y, *n* = 24) sjh (247 y, *n* = 25) xp (250 y, *n* = 25)			(O-Y)/Y
Contact	5.22 ± 2.67 c	3.36 ± 1.36 c	4.65 ± 3.02 c	9.62 ± 2.17 b	14.50 ± 2.83 ab	19.25 ± 2.79 a	2.38 ***
S_dist_coas	0.07 ± 0.02 c	0.03 ± 0.01 c	0.07 ± 0.03 bc	0.23 ± 0.06 b	0.46 ± 0.10 a	0.43 ± 0.07 a	5.92 ***
S_dist_dect	3.27 ± 1.03 c	7.32 ± 1.44 bc	3.51 ± 1.66 c	12.02 ± 1.99 ab	16.96 ± 2.43 a	13.91 ± 2.34 a	1.79 ***
M_dist_frng	0.40 ± 0.25 bc	1.17 ± 0.45 b	0.42 ± 0.34 c	0.75 ± 0.33 b	4.47 ± 2.78 a	4.76 ± 2.20 a	3.45 ***
M_dist_smth	1.89 ± 0.75 b	1.81 ± 1.12 b	2.41 ± 1.16 ab	2.19 ± 0.51 b	9.56 ± 2.64 a	7.51 ± 1.62 ab	2.29 ***
L_dist	0.03 ± 0.00 c	0.76 ± 0.44 ab	0.12 ± 0.04 bc	0.34 ± 0.16 b	0.52 ± 0.20 ab	0.46 ± 0.13 a	0.17 ***
L_dist/M_dist_frng ratio	0.52 ± 0.13 a	9.47 ± 7.74 a	4.08 ± 2.14 a	1.24 ± 0.48 a	0.69 ± 0.28 a	1.92 ± 1.11 a	−0.75 *
L_dist/M_dist_smth ratio	0.06 ± 0.02 b	5.10 ± 4.58 a	0.29 ± 0.12 ab	0.26 ± 0.12 ab	0.69 ± 0.51 b	0.15 ± 0.05 b	−0.85 *
L_dist/S_dist_dect ratio	0.04 ± 0.01 a	0.60 ± 0.46 a	0.12 ± 0.09 a	0.13 ± 0.10 a	0.04 ± 0.02 a	0.06 ± 0.02 a	NS
L_dist/S_dist_coas ratio	2.28 ± 0.83 b	41.29 ± 27.30 a	10.63 ± 4.99 ab	4.16 ± 1.64 b	2.28 ± 0.81 b	1.82 ± 0.58 b	−0.86 **
L_dist/Contact ratio	0.28 ± 0.09 ab	2.70 ± 2.13 a	5.70 ± 4.80 ab	0.15 ± 0.09 b	0.07 ± 0.03 b	0.03 ± 0.01 b	−0.96 ***
M_dist_frng/S_dist_dect ratio	0.16 ± 0.05 ab	0.30 ± 0.17 ab	0.08 ± 0.04 b	0.09 ± 0.03 b	0.50 ± 0.36 ab	1.59 ± 1.33 a	NS
M_dist_smth/S_dist_dect ratio	1.65 ± 0.74 a	0.37 ± 0.18 b	0.76 ± 0.42 ab	0.51 ± 0.25 ab	1.39 ± 0.54 ab	0.80 ± 0.16 a	NS
M_dist_frng/S_dist_coas ratio	7.74 ± 2.09 a	29.36 ± 7.22 a	10.39 ± 5.83 a	6.02 ± 1.46 a	18.09 ± 10.46 a	15.71 ± 5.26 a	NS
M_dist_smth/S_dist_coas ratio	58.50 ± 14.06 a	52.96 ± 17.56 a	242.37 ± 202.31 a	30.31 ± 10.28 a	45.30 ± 13.54 a	29.59 ± 6.69 a	NS
M_dist_frng/Contact ratio	12.80 ± 11.91 ab	3.60 ± 2.64 a	0.99 ± 0.37 ab	0.17 ± 0.04 b	3.37 ± 2.97 ab	0.47 ± 0.23 ab	−0.79 **
M_dist_smth/Contact ratio	0.71 ± 0.34 b	2.79 ± 1.07 ab	11.27 ± 6.29 a	9.58 ± 4.04 ab	4.01 ± 2.61 ab	0.65 ± 0.17 ab	−0.75 ***

**Table 3 jof-10-00310-t003:** Relative abundance of EcM fungal indicator taxa. The values are presented as the mean ± standard error. The different characters in a single row indicate significant differences across the six sites at *p* < 0.05. In the column of (O-Y)/Y, significant differences between the younger and older stands are denoted by asterisks as follows: *, *p* < 0.05; **, *p* < 0.01; and ***, *p* < 0.001; values denote increases (folds) from the older stands (O) to the younger stands (Y); NS indicates no significant difference between the younger and older stands. The differences are analyzed using one-way ANOVA. When the data do not meet the normality assumptions, they are analyzed using nonparametric Kruskal–Wallis tests. The exploration types include the contact (C), short-distance coarse (sdc), short-distance delicate (sdd), medium-distance fringe (mdf), medium-distance smooth (mds), and long-distance (L) types.

EcM Fungal Species (%)	Younger Stands lhs (114 y, *n* = 25) hnh (117 y, *n* = 25) ltd (122 y, *n* = 15)			Older Stands fh (251 y, *n* = 24) sjh (247 y, *n* = 25) xp (250 y, *n* = 25)			(O-Y)/Y
EcM18570.*Tomentella*49_mds	0 b	0.0044 ± 0.0012 a	0 b	0 b	0 b	0 b	-- ***
EcM347.*Tomentella*43_mds	0.0529 ± 0.0228 a	0 b	0 b	0 b	0.0002 ± 0.0002 b	0 b	−0.9971 ***
EcM15392.*Tomentella*11_mds	1.1771 ± 0.7551 a	0.0025 ± 0.0009 b	0.1035 ± 0.1010 b	0.0007 ± 0.0007 b	0.0119 ± 0.0119 b	0.0002 ± 0.0002 b	−0.9904 ***
EcM8957.*Cortinarius*35_mdf	0.0126 ± 0.0066 a	0.0012 ± 0.0005 b	0 c	0.0002 ± 0.0002 bc	0 c	0 c	−0.9889 ***
EcM8395.*Tomentella*5_mds	0.0220 ± 0.0128 a	0.0022 ± 0.0008 bc	0.0038 ± 0.0013 ab	0.0003 ± 0.0003 cd	0 d	0 d	−0.9883 ***
EcM18492.*Sebacina*27_sdd	0.0042 ± 0.0012 a	0.0037 ± 0.0014 ab	0.0027 ± 0.0012 ab	0.0005 ± 0.0003 bc	0.0002 ± 0.0002 c	0 c	−0.9390 ***
EcM19972.*Melanogaster*15_L	0.0023 ± 0.0010 bc	0.0673 ± 0.0228 a	0.0027 ± 0.0012 bc	0.0062 ± 0.0018 b	0.0002 ± 0.0002 c	0 c	−0.9324 ***
EcM11405.*Sebacina*42_sdd	0.0105 ± 0.0043 a	0.1205 ± 0.0638 a	0.0506 ± 0.0339 ab	0.0144 ± 0.0106 ab	0.0002 ± 0.0002 b	0 b	−0.9281 ***
EcM16292.*Sebacina*12_sdd	0.0174 ± 0.0098 bc	0.0634 ± 0.0220 a	0.0806 ± 0.0611 ab	0.0146 ± 0.0102 bc	0 c	0 c	−0.9036 ***
EcM17198.*Tomentella*63_mds	0.0031 ± 0.0011 a	0.0005 ± 0.0003 ab	0.0065 ± 0.0032 a	0.0005 ± 0.0005 b	0.0002 ± 0.0002 b	0.0002 ± 0.0002 b	−0.8933 ***
EcM17285.*Tomentella*51_mds	0.0042 ± 0.0018 a	0.0020 ± 0.0007 a	0.0034 ± 0.0019 a	0.0018 ± 0.0018 ab	0 b	0 b	−0.7997 ***
EcM5007.*Inocybe*56_sdd	0.0799 ± 0.0784 b	0 b	0 b	0.2271 ± 0.0971 a	0 b	0.0003 ± 0.0003 b	1.6563 *
EcM1607.*Sebacina*45_sdd	0 b	0.0003 ± 0.0003 b	0.0350 ± 0.0346 ab	0.0027 ± 0.0016 ab	0.0104 ± 0.0045 a	0.0618 ± 0.0336 a	2.5506 ***
EcM21424.*Tylospora*38_sdd	0.0013 ± 0.0009 cd	0.0097 ± 0.0050 cd	0.4655 ± 0.2663 c	0 d	0.4739 ± 0.2129 b	1.3498 ± 0.5243 a	5.3177 ***
EcM16453.*Hymenogaster*22_sdd	0.0063 ± 0.0050 bc	0.1819 ± 0.1715 b	0.0004 ± 0.0004 bc	2.2268 ± 0.7221 a	0.0003 ± 0.0003 c	0.0033 ± 0.0012 bc	7.9012 *
EcM12941.*Tomentella*28_mds	0.0002 ± 0.0002 c	0.0025 ± 0.0010 c	0.0346 ± 0.0244 bc	0.0395 ± 0.0218 bc	0.0740 ± 0.0299 ab	0.2308 ± 0.0731 a	13.354 ***
EcM18546.*Genea*7_sdc	0.0008 ± 0.0004 b	0.0028 ± 0.0014 b	0.0004 ± 0.0004 b	0.0141 ± 0.0040 a	0.0333 ± 0.0165 a	0.0236 ± 0.0098 a	13.390 ***
EcM12535.*Suillus*13_L	0.0086 ± 0.0019 ab	0.0027 ± 0.0008 b	0.0126 ± 0.0113 b	0.2850 ± 0.1577 a	0.0567 ± 0.0210 a	0.0571 ± 0.0303 a	18.770 ***
EcM23155.*Rhizopogon*1_L	0.0010 ± 0.0005 b	0.0028 ± 0.0007 b	0.0004 ± 0.0004 b	0.0161 ± 0.0048 a	0.0701 ± 0.0547 a	0.0594 ± 0.0338 a	27.243 ***
EcM13213.*Pseudotomentella*54_mds	0.0002 ± 0.0002 b	0.0012 ± 0.0009 b	0 b	0.0487 ± 0.0231 a	0.0037 ± 0.0027 b	0 b	28.213 *
EcM17847.*Otidea*25_sdc	0.0017 ± 0.0017 b	0 b	0 b	0.1026 ± 0.0396 a	0 b	0.0002 ± 0.0002 b	56.307 **
EcM20682.*Cenococcum*24_sdc	0 c	0.0022 ± 0.0012 c	0.0004 ± 0.0004 c	0.0015 ± 0.0010 c	0.1123 ± 0.0552 b	0.2003 ± 0.0642 a	99.107 ***
EcM12198.*Clavulina*46_C	0 b	0.0042 ± 0.0025 ab	0.0004 ± 0.0004 b	0.6249 ± 0.2894 a	0.0005 ± 0.0004 b	0.0136 ± 0.0125 b	NS
EcM5233.*Russula*36_C	0.0046 ± 0.0046 b	0 b	0 b	0.0062 ± 0.0051 b	0.1426 ± 0.0738 a	0.5200 ± 0.4635 a	134.62 ***
EcM18248.*Inocybe*29_sdd	0 b	0.0013 ± 0.0009 b	0 b	0.2408 ± 0.1243 a	0.1948 ± 0.1099 a	0.0072 ± 0.0059 b	245.96 ***
EcM14251.*Sebacina*64_sdd	0 b	0.0003 ± 0.0002 b	0 b	0 b	0.0549 ± 0.0329 a	0.1630 ± 0.0656 a	485.08 ***
EcM17329.*Inocybe*2_sdd	0.0008 ± 0.0007 c	0.0020 ± 0.0017 c	0 c	1.3903 ± 0.5021 a	0.4446 ± 0.2207 ab	0.0351 ± 0.0217 b	520.50 ***
EcM10960.*Lactifluus*57_mds	0.0046 ± 0.0032 b	0 b	0 b	0 b	2.0139 ± 1.3653 a	1.0093 ± 0.6139 a	612.12 ***
EcM11301.*Russula*3_C	0.004 ± 0.003 cd	0 d	0 d	0.0649 ± 0.0405 bc	0.0619 ± 0.0375 ab	0.2107 ± 0.0823 a	752.01 ***
EcM15576.*Russula*23_C	0 b	0.0002 ± 0.0002 b	0 b	0.0202 ± 0.0164 ab	0.0308 ± 0.0210 a	0.1279 ± 0.0658 a	797.19 ***
EcM10920.*Lactifluus*50_mds	0.0023 ± 0.0017 b	0.0002 ± 0.0002 b	0 b	0 b	2.0095 ± 1.3214 a	0.2624 ± 0.1404 a	843.73 ***
EcM10378.*Sebacina*9_sdd	0.0008 ± 0.0005 c	0.0002 ± 0.0002 c	0 c	0.0002 ± 0.0002 c	0.6563 ± 0.2533 b	0.8736 ± 0.2801 a	1364.4 ***
EcM10234.*Pseudocraterellus*17_sdd	0.0002 ± 0.0002 b	0 b	0 b	0 b	0.2504 ± 0.1733 a	0.2015 ± 0.1003 a	2015 ***
EcM10122.*Lactifluus*26_mds	0.0002 ± 0.0002 c	0 c	0 c	0 c	0.3946 ± 0.2117 a	0.3635 ± 0.2125 b	3381.4 ***
EcM16609.*Wilcoxina*8_sdc	0 b	0 b	0 b	0 b	0.0758 ± 0.0287 a	0.0592 ± 0.0207 a	-- ***
EcM16354.*Tomentella*44_mds	0 b	0 b	0 b	0.0002 ± 0.0002 b	0.0008 ± 0.0005 b	0.2678 ± 0.1013 a	-- ***
EcM17223.*Tomentella*39_mds	0 b	0 b	0 b	0.0142 ± 0.0080 a	0 b	0.0002 ± 0.0002 b	-- ***
EcM17687.*Russula*37_C	0 b	0 b	0 b	0.0013 ± 0.0008 b	0.0005 ± 0.0004 b	0.1561 ± 0.0785 a	-- ***
EcM22874.*Piloderma*41_mdf	0 b	0 b	0 b	0.0003 ± 0.0003 b	0.1884 ± 0.1180 a	0.0515 ± 0.0178 a	-- ***
EcM18178.*Genea*31_sdc	0 c	0 c	0 c	0.0328 ± 0.0145 b	0.0695 ± 0.0353 a	0 c	-- ***
EcM11003.*Clavulina*6_C	0 b	0 b	0 b	0 b	0.3262 ± 0.1570 a	0.9691 ± 0.3412 a	-- ***
EcM16726.*Amanita*52_C_mds	0 b	0 b	0 b	0.0370 ± 0.0154 a	0.0639 ± 0.0486 ab	0.0030 ± 0.0019 ab	-- ***

## Data Availability

Dataset and associated R codes used in the main results are available upon reasonable request to the corresponding author.

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
