# Peer review of "Age-Related Conservation in Plant–Soil Feedback Accompanied by Ectomycorrhizal Domination in Temperate Forests in Northeast China"

_jof, 2024, doi:10.3390/jof10050310_

Round 1
Reviewer 1 Report
Dear authors, I hope the following observations are useful to you:
The manuscript talks about the dominance of ectomycorrhizal fungi in temperate forests. I consider that another approach should be taken to the work, since the samples used are soil samples at a maximum depth of 10 cm and were taken randomly and mixed.
What is the life time of this type of forest in natural conditions?. This is to be able to mention that young is ~120 years and older is ~250 years.
By taking the samples at a maximum depth of 10 cm in the soil, it is not known if these fungi are in interaction with the forest trees; they would have to have been taken at a greater depth and in the rhizosphere of the trees.
The mixtures of the samples only reveal the possible majority population of ectomycorrhizal fungi. However, this does not mean that these fungi are associated with trees; it is possible that they only degrade dead plant material on the forest floor.
The article focuses a lot on the characterization of trees, so perhaps the manuscript is a better candidate for another journal like Forest.
none
Author Response
Major comments
The manuscript talks about the dominance of ectomycorrhizal fungi in temperate forests. I consider that another approach should be taken to the work, since the samples used are soil samples at a maximum depth of 10 cm and were taken randomly and mixed. What is the life time of this type of forest in natural conditions? This is to be able to mention that young is ~120 years and older is ~250 years. By taking the samples at a maximum depth of 10 cm in the soil, it is not known if these fungi are in interaction with the forest trees; they would have to have been taken at a greater depth and in the rhizosphere of the trees. The mixtures of the samples only reveal the possible majority population of ectomycorrhizal fungi. However, this does not mean that these fungi are associated with trees; it is possible that they only degrade dead plant material on the forest floor. The article focuses a lot on the characterization of trees, so perhaps the manuscript is a better candidate for another journal like Forest.
Reply: Many thanks. We have thoroughly revised the contents in this manuscript (Please see the manuscript with changed marks). Temperature forest ecosystem is highly dependent upon ectomycorrhizal (EcM) fungi. In particular, the plant–soil attributes in temperate forests are highly associated with EcM fungi in soil. However, little is known about the effects of temperate forest aging on EcM fungal community and foraging behavior and the consequent influences on plant–soil attributes. This study explored EcM fungal composition and hyphal exploration types and specific indicator taxa by sequencing fungal 18S rDNA with Illumina Miseq, and investigated their associations with plant–soil traits, e.g., leaf nutrient contents, tree height, annual aboveground biomass increment indicated by coarse woody productivity, and soil nutrient status, by comparing younger (~120 years) and older (~250 years) stands in temperate forests in Northeast China. Our findings underscore the integral role of soil EcM fungi in the aging processes of temperate forests, highlighting the EcM symbiont-mediated mechanisms adapting to soil nutrient scarcity and promoting sustainability in plant–soil consortia.
Reviewer 2 Report
The present works analyze the differencies in young and old growing forests regarding ECM community and variables of forest stands. Moreover , analysis of exploration strategy suggest a specific selection of ECM fungal taxa according to forest age. I think this is a good piece of science and this work can contribute to knowledge on the topic. Reading was good but in some point difficult as main topic is lost. Additionally some terms like plant soil continuum should be avoided as do not substantially contribute to the quality of the work. Main problems arise from statistics and figure quality. Statistic methodologies are good in some points but require rationale. declaring Normality test without showing the results is inappropriate. Figures could be arranged differently without using abbreviations facilitating reagings and interpretations. Further specific comments can be found below. Finally, please revise bibliography as many citation seems to be poorly appropriate.
Abstract:
l18, not needed to declare primer and DNA region. If the authors think is necessary, please provide extensive name.
l19-25, what the authors means by woody production/productivity?
L28 already mentioned this result in line 22, this part of the abstract should be reserved for result implications regarding the topic.
Abstract should be rephrased, may be clearer for readers if the authors briefly state hypotheses. Provide salient results and explanation.
Introduction:
Citation number 8: inappropriate
L43… sequestration of nutrients by soil…
L57 consider to add citation: https://pubmed.ncbi.nlm.nih.gov/37374896/
L57-69 the authors here need to be more concise. I think this part could be synthesized and concentrated on accumulation of ECM fungal species in old growing forests compared to young forests. I suggest the author to consider publication on Gadgil effect hypothesis. Moreover, direction of plant soil feedback when considering ECM tree dominated ecosystems is normally positive in broadleaved forest, while variable if not negative in Coniferous dominated forest. This effect is related to renewal on forest stands and formation of plant community in successional stages. To the best of what I understand this work did not measure plant soil feedback and relations of ECM community with renewal. I suggest to focalize the attention on ECM community composition in relation with Forest age and functionality.
L70 start by Ecm fungi,……
L70 instead of integral use : are associated to …… or are related to vegetation aging, woody plant community, ……
L76 this is a clear case of Gadgil effect. Please consider substituting the present sentence in a new form: established ECM community exclude fungal saprotrophs monopolizing organic matter decomposition and transferring that to below ground soil matrix.
L78 tissues is totally inappropriate for fungi.
L89 please add citation
L98 citation 29 refer to a chrono-sequence in a single stand, successional stages are basically formed by different composition of forest stands. The definition of early and later successional stage is basically inappropriate. Just consider forest aging.
L100 Wath citation 28 states? it seems that citation of Santiago 2008 treats about Eucalyptus pulping rather than fungal community associated to plant aging. Same for citation 30 and others.
L112 wrong reference style please revise here and throughout the manuscript.
L 123 this hypothesis is redundant with respect to the first.
L125 keystone is inappropriate, Keystone taxa refers to species that are pivotal for ecosystem stability and existence, without them the ecosystem will collapse. The authors should use the term indicator species as this work is not based on a taxa removal experiment.
Materials and methods
L194 how taxonomy was assigned? UNITE? NCBI? BOLD?
L194 where sequences of amplicon study were deposited?
L196 do the authors performed levene test to check for homogeneity of variance? Do the author perform data transformation before assuming to use non parametric test ? additionally, Hypothesys testing of Anova/Kruskal-Wallis should be declared here.
L201The author used pearson correlation for all the variables tested ? in case of non normal distribution separate spearman rank coefficient should be calculated. I suggest using directly the non parametric option in case of non-normal distribution of the data.
L203 The rationale behind nmds should be rephrased. May be the difference in ECM community composition in relation to stand age.
L206 add description of permanova design. Moreover move result of permanova test in appropriate section.
Before multivariate analyses has data been transformed? Do the authors used Raw reads frequencies ? or standardized ?
Results
L215-226 so the plant community is different with few ecm tree species in young forest rather than in old forest. I suggest putting this part at the end of the introduction section evidencing the different successional stage and motivating it. From how the introduction were written it appears that forest with same composition but with different ages were compared.
Figure 1 please avoid abbreviations as variable labels. Codes makes difficult understandability of the figures.
Table 1 caption inacceptable. How was significance calculated? Anova, Kruskal-Wallis?
In figures concerning variation rates. Why are them so high in old forests? (15000%) maybe further transformation of the data is required.
Figure 2, please provide number of permutation, legnd is not clearly visible and please use tree names.
Table 2 as table 1, also in this case caption is inconsistent. I suggest to move table footnotes in caption section.
Figure 3 caption should be self-explanatory of the figure. correlation method should be added. I think that there is enough space to put extended names of variables. Putting extended names allow to avoid listing of variables in caption. I see asterisks, please add explanation of asterisks meaning, if significance is assigned explain why.
Figure 4, did the author considered to make a Vulcan plot for comparison of two sample group? Variance issues are solved by log fold changes transformation.
Figure 5 use extended names for variables (at least for tree species) and specify correlation method.
Table 3 revise caption

Author Response
Reviewer2
Major comments
- M1: The present works analyze the differencies in young and old growing forests regarding ECM community and variables of forest stands. Moreover , analysis of exploration strategy suggest a specific selection of ECM fungal taxa according to forest age. I think this is a good piece of science and this work can contribute to knowledge on the topic.
Reply: Many thanks for your nice comments! We have carefully revised the contents as you mentioned below. We hope you are satisfied with our replies and the revised contents in the manuscript.
- M2: Reading was good but in some point difficult as main topic is lost.
Reply: Many thanks. We have thoroughly revised the contents in this manuscript (Please see the manuscript with changed marks).
- M3: Additionally some terms like plant soil continuum should be avoided as do not substantially contribute to the quality of the work.
Reply: Many thanks and done.
- M4: Main problems arise from statistics and figure quality. Statistic methodologies are good in some points but require rationale. 1) Declaring Normality test without showing the results is inappropriate. 2) Figures could be arranged differently without using abbreviations facilitating readings and interpretations.
Reply: Many thanks.
1) We revised the contents related to statistics as follows: “The influences of forest aging on EcM fungal alpha-diversity, exploration types and indicator species were evaluated as follows. The normality and the variance homogeneity of the data were assessed using the Shapiro–Wilk and Levene’s tests, respectrively. For data that were normally distributed, differences were analyzed using one-way analysis of variance (ANOVA) with post-hoc comparisons via Tukey’s honestly significant difference (HSD) test. Data not metting the normality assumption were analyzed using the Kruskal–Wallis test, a nonparametric alternative. Raw data was standardized before further analyses. The analyses were executed using the statistical functions available in R software (R Development Core Team, 2022, version 4.1.3). Statistical significance was defined as P < 0.05. We performed a spearman’s correlation analysis on variables to assess correlations via the ‘corrr’ package in R. We carried out nonmetric multidimensional scaling analysis (NMDS) to explore the difference in EcM fungal community in relation to forest stand age. Analysis of similarity (ANOSIM) and permutational multivariate analysis of variance (PERMANOVA) revealed the influence of aging effect with Bray–Curtis similarities and 999 permutations. PERMANOVA relies on permutation tests and can be used to data not fitting the normatlity assumption [56].”
2) We replaced the abbreviations by their full names, and revised all figures in this manuscript.
- M5: Finally, please revise bibliography as many citation seems to be poorly appropriate.
Reply: Many thanks and done. We thoroughly checked the citations in this mansucirpt.
Detail comments
Abstract:
- D1: l18, not needed to declare primer and DNA region. If the authors think is necessary, please provide extensive name.
Reply: Many thanks and done. The content was revised as “sequencing fungal 18S rDNA with Illumina Miseq”,
- D2: l19-25, what the authors means by woody production/productivity?
Reply: Many thanks. The contents were rephrased as “annual aboveground biomass increment indicated by coarse woody productivity”.
- D3: L28 already mentioned this result in line 22, this part of the abstract should be reserved for result implications regarding the topic.
Reply: Many thanks. These contents were separately rephrased as 1) “paralleled by plant–soil feedbacks shifting from explorative to conservative nutrient use strategies with forest aging”; 2) “Notably, the ratios of longer-to-shorter EcM fungal exploration distance types tended to decrease along forest aging.”
- D4: Abstract should be rephrased, may be clearer for readers if the authors briefly state hypotheses. Provide salient results and explanation.
Reply: Many thanks. We had revised the abstract thoroughly.
Introduction:
- D5: Citation number 8: inappropriate
Reply: Many thanks and done. At L53, we removed the wrong citations and added new and closely-related references according to the context.
- D6: L43… sequestration of nutrients by soil…
Reply: Many thanks and done.
- D7: L57 consider to add citation: https://pubmed.ncbi.nlm.nih.gov/37374896/
Reply: Many thanks and done.
- D8: L57-69 the authors here need to be more concise. I think this part could be synthesized and concentrated on accumulation of ECM fungal species in old growing forests compared to young forests. I suggest the author to consider publication on Gadgil effect hypothesis. Moreover, direction of plant soil feedback when considering ECM tree dominated ecosystems is normally positive in broadleaved forest, while variable if not negative in Coniferous dominated forest. This effect is related to renewal on forest stands and formation of plant community in successional stages. To the best of what I understand this work did not measure plant soil feedback and relations of ECM community with renewal. I suggest to focalize the attention on ECM community composition in relation with Forest age and functionality.
Reply: Many thanks. The contents had been revised, “Thus, forest aging is believed to result in composition and functional convergence within soil microbial communities[12,13]. Nonetheless, the mechanisms underlying plant–soil–fungal dynamics and related nutrient use strategies remain largely ambiguous as forests progress through different age stages.”
- D9: L70 start by Ecm fungi,……
Reply: Many thanks and done.
- D10: L70 instead of integral use : are associated to …… or are related to vegetation aging, woody plant community, ……
Reply: Many thanks. The contents were revised as “EcM fungi are closely related to vegetation community assembly and soil nutrient cycling dynamics during forest aging”.
- D11: L76 this is a clear case of Gadgil effect. Please consider substituting the present sentence in a new form: established ECM community exclude fungal saprotrophs monopolizing organic matter decomposition and transferring that to below ground soil matrix.
Reply: Many thanks and done.
- D12: L78 tissues is totally inappropriate for fungi.
Reply: Many thanks. We replaced “tissues” by “extraradical mycelia”.
- D13: L89 please add citation
Reply: Many thanks and done.
- D14: L98 citation 29 refer to a chrono-sequence in a single stand, successional stages are basically formed by different composition of forest stands. The definition of early and later successional stage is basically inappropriate. Just consider forest aging.
Reply: Many thanks. We removed the previously cited citation and added a new citation closely related to forest aging, “For instance, early successional-stage forests are highly abundant with cord-forming EcM basidiomycetes (Suillus and Piloderma spp.).”
(Citation: Clemmensen, K.E.; Finlay, R.D.; Dahlberg, A.; Stenlid, J.; Wardle, D.A.; Lindahl, B.D. Carbon sequestration is related to mycorrhizal fungal community shifts during long-term succession in boreal forests. New Phytol. 2015, 205, 1525-1536, doi:10.1111/nph.13208.)
- D15: L100 Wath citation 28 states? it seems that citation of Santiago 2008 treats about Eucalyptus pulping rather than fungal community associated to plant aging. Same for citation 30 and others.
Reply: Many thanks! We thoroughly checked and revised citations in this manuscript.
- D16: L112 wrong reference style please revise here and throughout the manuscript.
Reply: Many thanks and done.
- D17: L 123 this hypothesis is redundant with respect to the first.
Reply: Many thanks. We revised the second hypothesis, “2) a shift in EcM fungal community composition, potentially affecting the balance between long- and short-distance types;”
- D18: L125 keystone is inappropriate, Keystone taxa refers to species that are pivotal for ecosystem stability and existence, without them the ecosystem will collapse. The authors should use the term indicator species as this work is not based on a taxa removal experiment.
Reply: Many thanks. We revised the third hypothesis, “3) the emergence of indicator taxa of EcM fungi specific to different forest successional stages”. We removed “keystone” in this manuscript.
Materials and methods
- D19: L194 how taxonomy was assigned? UNITE? NCBI? BOLD?
Reply: Many thanks. We added this information, “We assigned taxonomy with representative sequences from UNITE7.0 database.”
- D20: L194 where sequences of amplicon study were deposited?
Reply: Many thanks. This work will be finished soon.
- D21: L196 do the authors performed levene test to check for homogeneity of variance? Do the author perform data transformation before assuming to use non parametric test ? additionally, Hypothesys testing of Anova/Kruskal-Wallis should be declared here.
Reply: Many thanks. The contents were revised as follows: “The influences of forest aging on EcM fungal alpha-diversity, exploration types and indicator species were evaluated as follows. The normality and the variance homogeneity of the data were assessed using the Shapiro–Wilk and Levene’s tests, respectrively. For data that were normally distributed, differences were analyzed using one-way analysis of variance (ANOVA) with post-hoc comparisons via Tukey’s honestly significant difference (HSD) test. Data not metting the normality assumption were analyzed using the Kruskal–Wallis test, a nonparametric alternative. Raw data was standardized before further analyses. The analyses were executed using the statistical functions available in R software (R Development Core Team, 2022, version 4.1.3). Statistical significance was defined as P < 0.05. We performed a spearman’s correlation analysis on variables to assess correlations via the ‘corrr’ package in R. We carried out nonmetric multidimensional scaling analysis (NMDS) to explore the difference in EcM fungal community in relation to forest stand age. Analysis of similarity (ANOSIM) and permutational multivariate analysis of variance (PERMANOVA) revealed the influence of aging effect with Bray–Curtis similarities and 999 permutations. PERMANOVA relies on permutation tests and can be used to data not fitting the normatlity assumption [56].”
- D22: L201The author used pearson correlation for all the variables tested ? in case of non normal distribution separate spearman rank coefficient should be calculated. I suggest using directly the non parametric option in case of non-normal distribution of the data.
Reply: Many thanks and done. We performed a spearman’s correlation analysis on variables to assess correlations via the ‘corrr’ package in R.
- D23: L203 The rationale behind nmds should be rephrased. May be the difference in ECM community composition in relation to stand age.
Reply: Many thanks and done.
- D24: L206 add description of permanova design. Moreover move result of permanova test in appropriate section.
Reply: Many thanks and done.
- D25: Before multivariate analyses has data been transformed? Do the authors used Raw reads frequencies ? or standardized ?
Reply: Thanks. We standardized the data before further analysis
Results
- D26: L215-226 so the plant community is different with few ecm tree species in young forest rather than in old forest. I suggest putting this part at the end of the introduction section evidencing the different successional stage and motivating it. From how the introduction were written it appears that forest with same composition but with different ages were compared.
Reply: Many thanks and done.
- D27: Figure 1 please avoid abbreviations as variable labels. Codes makes difficult understandability of the figures.
Reply: Many thanks and done.
- D28: Table 1 caption inacceptable. How was significance calculated? Anova, Kruskal-Wallis?
Reply: Many thanks and done.
- D29: In figures concerning variation rates. Why are them so high in old forests? (15000%) maybe further transformation of the data is required.
Reply: Many thanks and done.
- D30: Figure 2, please provide number of permutation, legnd is not clearly visible and please use tree names.
Reply: Many thanks and done.
- D31: Table 2 as table 1, also in this case caption is inconsistent. I suggest to move table footnotes in caption section.
Reply: Many thanks and done.
- D32: Figure 3 caption should be self-explanatory of the figure. correlation method should be added. I think that there is enough space to put extended names of variables. Putting extended names allow to avoid listing of variables in caption. I see asterisks, please add explanation of asterisks meaning, if significance is assigned explain why.
Reply: Many thanks and done.
- D33: Figure 4, did the author considered to make a Vulcan plot for comparison of two sample group? Variance issues are solved by log fold changes transformation.
Reply: Many thanks for your nice comments. We will try a Vulcan plot in the future on the data of EcM fungal species.
- D34: Figure 5 use extended names for variables (at least for tree species) and specify correlation method.
Reply: Many thanks and done.
- D35: Table 3 revise caption
Reply: Many thanks and done.
Round 2
Reviewer 1 Report
Dear Authors,
The second version of the uploaded manuscript does not have the marks on what have modified. Also requested you in the most attentive manner, to respond point by point to the observations made in the first review.
None
Reviewer 2 Report
The author make a good effort to adress all the comments. The quality of the work significally improved.
The author make a good effort to adress all the comments. The quality of the work significally improved.
Round 3
Reviewer 1 Report
My only observation is regarding the nomenclature of Illumina sequencing. Since in some paragraphs they mention that the sequencing analysis was of the 18S rDNA region and in others the ITS region. They must correct this since it creates confusion, in materials and methods they use oligonucleotides for the ITS region, so I would consider this to be the correct one. I also suggest that the data obtained from Illumina sequencing be uploaded to some database.
My only observation is regarding the nomenclature of Illumina sequencing. Since in some paragraphs they mention that the sequencing analysis was of the 18S rDNA region and in others the ITS region. They must correct this since it creates confusion, in materials and methods they use oligonucleotides for the ITS region, so I would consider this to be the correct one. I also suggest that the data obtained from Illumina sequencing be uploaded to some database.
Author Response
Dear editors,
We hereby submit our manuscript entitled “Age-related conservation in plant–soil feedbacks accompanied by ectomycorrhizal domination in temperate forests, Northeast China,” which we wish to be considered for publication in Journal of Fungi. This is a resubmission of Ms. Ref. No.: jof-2907954, which is previously handled by Editor Silvia Alexandra Avram. A point-by-point response to the comments by the editor and reviewers had been attached (please see below).
If you need further information, please let us know. We greatly appreciate your consideration of this manuscript.
Sincerely,
Zhen Bai, Ph.D.
April, 11th 2024
E-mail addresses: baizhen@iae.ac.cn (Z. Bai).
Editor
Major comments
- Please check that all references are relevant to the contents of the manuscript.
Reply: Many thanks. We have thoroughly checked all references and ensure their relevance to the contents of the manuscript. The references and related contents were revised at lines 155-156 [ref 47, 48], 157-159 [ref 45, 46, 49], 199-200 [ref 52], 475-477 [ref 54, 55], 491-496 [ref 6, 7, 10, 16, 31], 497-499 [ref 31, 60], 519-522 [ref 16, 29, 31, 64], 526-530 [ref 29, 66, 67], 541-545 [ref 72], 546-550 [ref 73, 74, 29, 75], 563-564 [ref 33, 76-77], 586-591 [ref [25, 33, 77, 82-84], 591-593 [ref 85], 596-597 [ref 25, 33, 76], 597-600 [ref 21, 29, 88, 89], and 606-609 [ref 85, 91].
Reviewer 1
Major comments
- My only observation is regarding the nomenclature of Illumina sequencing. Since in some paragraphs they mention that the sequencing analysis was of the 18S rDNA region and in others the ITS region. They must correct this since it creates confusion, in materials and methods they use oligonucleotides for the ITS region, so I would consider this to be the correct one.
Reply: Many thanks and done! We have corrected the contents and removed “18S” at lines 23, 179, 192 and 194.
- I also suggest that the data obtained from Illumina sequencing be uploaded to some database.
Reply: Many thanks and done. We have added the contents at lines 223-224, “All raw sequence reads are archived at the NCBI sequence Read Archive under accession number SUB14383148”.